# Legendre Decomposition for Tensors

**Mahito Sugiyama**
National Institute of Informatics
JST, PRESTO
mahito@nii.ac.jp

**Hiroyuki Nakahara**
RIKEN Center for Brain Science
hiro@brain.riken.jp

**Koji Tsuda**
The University of Tokyo
NIMS; RIKEN AIP
tsuda@k.u-tokyo.ac.jp

## Abstract

We present a novel *nonnegative tensor decomposition* method, called *Legendre decomposition*, which factorizes an input tensor into a multiplicative combination of parameters. Thanks to the well-developed theory of information geometry, the reconstructed tensor is unique and always minimizes the KL divergence from an input tensor. We empirically show that Legendre decomposition can more accurately reconstruct tensors than other nonnegative tensor decomposition methods.

## 1 Introduction

Matrix and tensor decomposition is a fundamental technique in machine learning; it is used to analyze data represented in the form of multi-dimensional arrays, and is used in a wide range of applications such as computer vision (Vasilescu and Terzopoulos, 2002, 2007), recommender systems (Symeonidis, 2016), signal processing (Cichocki et al., 2015), and neuroscience (Beckmann and Smith, 2005). The current standard approaches include nonnegative matrix factorization (NMF; Lee and Seung, 1999, 2001) for matrices and CANDECOMP/PARAFAC (CP) decomposition (Harshman, 1970) or Tucker decomposition (Tucker, 1966) for tensors. CP decomposition compresses an input tensor into a sum of rank-one components, and Tucker decomposition approximates an input tensor by a core tensor multiplied by matrices. To date, matrix and tensor decomposition has been extensively analyzed, and there are a number of variations of such decomposition (Kolda and Bader, 2009), where the common goal is to approximate a given tensor by a smaller number of components, or parameters, in an efficient manner.

However, despite the recent advances of decomposition techniques, a learning theory that can systematically define decomposition for *any order tensors* including vectors and matrices is still under development. Moreover, it is well known that CP and Tucker tensor decomposition include non-convex optimization and that the global convergence is not guaranteed. Although there are a number of extensions to transform the problem into a convex problem (Liu et al., 2013; Tomioka and Suzuki, 2013), one needs additional assumptions on data, such as a bounded variance.

Here we present a new paradigm of matrix and tensor decomposition, called *Legendre decomposition*, based on *information geometry* (Amari, 2016) to solve the above open problems of matrix and tensor decomposition. In our formulation, an arbitrary order tensor is treated as a discrete probability distribution in a statistical manifold as long as it is nonnegative, and Legendre decomposition is realized as a *projection* of the input tensor onto a submanifold composed of reconstructable tensors. The key to introducing the formulation is to use the *partial order* (Davey and Priestley, 2002; Gierz et al., 2003) of indices, which allows us to treat any order tensors as a probability distribution in the information geometric framework.

Legendre decomposition has the following remarkable properties: It always finds the unique tensor that minimizes the Kullback–Leibler (KL) divergence from an input tensor. This is because Legendre decomposition is formulated as convex optimization, and hence we can directly use *gradient descent*, which always guarantees the global convergence, and the optimization can be further speeded up by

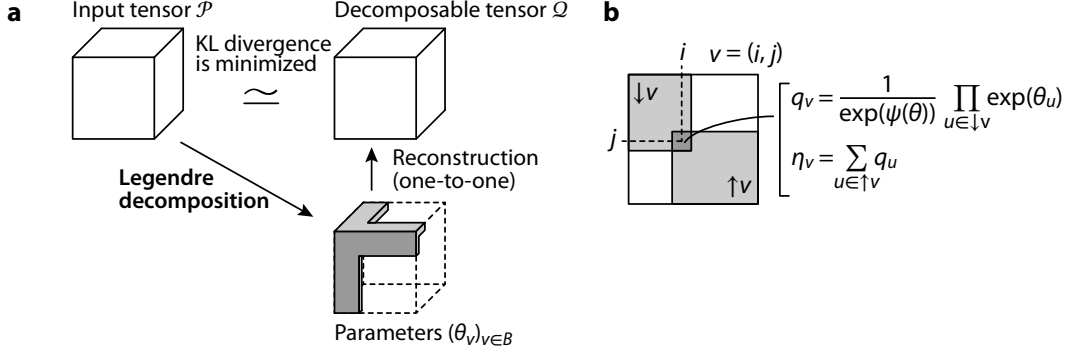

Figure 1: (**a**) Overview of Legendre decomposition. (**b**) Illustration of $\theta$ and $\eta$ for second-order tensor (matrix) when $B = [I_1] \times [I_2]$.

using a *natural gradient* (Amari, 1998) as demonstrated in our experiments. Moreover, Legendre decomposition is flexible as it can decompose sparse tensors by removing arbitrary entries beforehand, for examples zeros or missing entries.

Furthermore, our formulation has a close relationship with statistical models, and can be interpreted as an extension of the learning of *Boltzmann machines* (Ackley et al., 1985). This interpretation gives new insight into the relationship between tensor decomposition and graphical models (Chen et al., 2018; Yılmaz et al., 2011; Yılmaz and Cemgil, 2012) as well as the relationship between tensor decomposition and energy-based models (LeCun et al., 2007). In addition, we show that the proposed formulation belongs to the *exponential family*, where the parameter $\theta$ used in our decomposition is the natural parameter, and $\eta$, used to obtain the gradient of $\theta$, is the expectation of the exponential family.

The remainder of this paper is organized as follows. We introduce Legendre decomposition in Section 2. We define the decomposition in Section 2.1, formulate it as convex optimization in Section 2.2, describe algorithms in Section 2.3, and discuss the relationship with other statistical models in Section 2.4. We empirically examine performance of our method in Section 3, and summarize our contribution in Section 4.

## 2   The Legendre Decomposition

We introduce *Legendre decomposition* for tensors. We begin with a nonnegative $N$th-order tensor $\mathcal{X} \in \mathbb{R}_{\geq 0}^{I_1 \times I_2 \times \cdots \times I_N}$. To simplify the notation, we write the entry $x_{i_1 i_2 \dots i_N}$ by $x_v$ with the index vector $v = (i_1, i_2, \dots, i_N) \in [I_1] \times [I_2] \times \cdots \times [I_N]$, where each $[I_k] = \{1, 2, \dots, I_k\}$. To treat $\mathcal{X}$ as a discrete probability mass function in our formulation, we normalize $\mathcal{X}$ by dividing each entry by the sum of all entries, yielding $\mathcal{P} = \mathcal{X} / \sum_v x_v$. In the following, we always work with a normalized tensor $\mathcal{P}$.

### 2.1   Definition

Legendre decomposition always finds the best approximation of a given tensor $\mathcal{P}$. Our strategy is to choose an index set $B \subseteq [I_1] \times [I_2] \times \cdots \times [I_N]$ as a *decomposition basis*, where we assume $(1, 1, \dots, 1) \notin B$ for a technical reason, and approximate the normalized tensor $\mathcal{P}$ by a multiplicative combination of parameters associated with $B$.

First we introduce the relation "$\leq$" between index vectors $u = (j_1, \dots, j_N)$ and $v = (i_1, \dots, i_N)$ as $u \leq v$ if $j_1 \leq i_1, j_2 \leq i_2, \dots, j_N \leq i_N$. It is clear that this relation gives a *partial order* (Gierz et al., 2003); that is, $\leq$ satisfies the following three properties for all $u, v, w$: (1) $v \leq v$ (reflexivity), (2) $u \leq v, v \leq u \Rightarrow u = v$ (antisymmetry), and (3) $u \leq v, v \leq w \Rightarrow u \leq w$ (transitivity). Each tensor is treated as a discrete probability mass function with the sample space $\Omega \subseteq [I_1] \times \cdots \times [I_N]$. While it is natural to set $\Omega = [I_1] \times \cdots \times [I_N]$, our formulation allows us to use any subset $\Omega \subseteq [I_1] \times \cdots \times [I_N]$. Hence, for example, we can remove unnecessary indices such as missing or zero entries of an input tensor $\mathcal{P}$ from $\Omega$. We define $\Omega^+ = \Omega \setminus \{(1, 1, \dots, 1)\}$.

We define a tensor $\mathcal{Q} \in \mathbb{R}_{\geq 0}^{I_1 \times I_2 \times \cdots \times I_N}$ as *fully decomposable with $B \subseteq \Omega^+$* if each entry $q_v \in \Omega$ is represented in the form of

$$q_v = \frac{1}{\exp(\psi(\theta))} \prod_{u \in \downarrow v} \exp(\theta_u), \quad \downarrow v = \{\, u \in B \mid u \leq v \,\}, \tag{1}$$

using $|B|$ parameters $(\theta_v)_{v \in B}$ with $\theta_v \in \mathbb{R}$ and the normalizer $\psi(\theta) \in \mathbb{R}$, which is always uniquely determined from the parameters $(\theta_v)_{v \in B}$ as

$$\psi(\theta) = \log \sum_{v \in \Omega} \prod_{u \in \downarrow v} \exp(\theta_u).$$

This normalization does not have any effect on the decomposition performance, but rather it is needed to formulate our decomposition as an information geometric projection, as shown in the next subsection. There are two extreme cases for a choice of a basis $B$: If $B = \emptyset$, a fully decomposable $\mathcal{Q}$ is always uniform; that is, $q_v = 1/|\Omega|$ for all $v \in \Omega$. In contrast, if $B = \Omega^+$, any input $\mathcal{P}$ itself becomes decomposable.

We now define *Legendre decomposition* as follows: Given an input tensor $\mathcal{P} \in \mathbb{R}_{\geq 0}^{I_1 \times I_2 \times \cdots \times I_N}$, the sample space $\Omega \subseteq [I_1] \times [I_2] \times \cdots \times [I_N]$, and a parameter basis $B \subseteq \Omega^+$, Legendre decomposition finds the fully decomposable tensor $\mathcal{Q} \in \mathbb{R}_{\geq 0}^{I_1 \times I_2 \times \cdots \times I_N}$ with a $B$ that minimizes the Kullback–Leibler (KL) divergence $D_{\mathrm{KL}}(\mathcal{P}, \mathcal{Q}) = \sum_{v \in \Omega} p_v \log(p_v/q_v)$ (Figure 1[a]). In the next subsection, we introduce an additional parameter $(\eta_v)_{v \in B}$ and show that this decomposition is always possible via the dual parameters $(\,(\theta_v)_{v \in B}, (\eta_v)_{v \in B}\,)$ with information geometric analysis. Since $\theta$ and $\eta$ are connected via *Legendre transformation*, we call our method Legendre decomposition.

Legendre decomposition for second-order tensors (that is, matrices) can be viewed as a low rank approximation not of an input matrix $\mathcal{P}$ but of its entry-wise logarithm $\log \mathcal{P}$. This is why the rank of $\log \mathcal{Q}$ with the fully decomposable matrix $\mathcal{Q}$ coincides with the parameter matrix $\mathcal{T} \in \mathbb{R}^{I_1 \times I_2}$ such that $t_v = \theta_v$ if $v \in B$, $t_{(1,1)} = 1/\exp(\psi(\theta))$, and $t_v = 0$ otherwise. Thus we fill zeros into entries in $\Omega^+ \setminus B$. Then we have $\log q_v = \sum_{u \in \downarrow v} t_u$, meaning that the rank of $\log \mathcal{Q}$ coincides with the rank of $\mathcal{T}$. Therefore if we use a decomposition basis $B$ that includes only $l$ rows (or columns), $\mathrm{rank}(\log \mathcal{Q}) \leq l$ always holds.

## 2.2 Optimization

We solve the Legendre decomposition by formulating it as a convex optimization problem. Let us assume that $B = \Omega^+ = \Omega \setminus \{(1,1,\ldots,1)\}$, which means that any tensor is fully decomposable. Our definition in Equation (1) can be re-written as

$$\log q_v = \sum_{u \in \Omega^+} \zeta(u,v)\theta_u - \psi(\theta) = \sum_{u \in \Omega} \zeta(u,v)\theta_u, \quad \zeta(u,v) = \begin{cases} 1 & \text{if } u \leq v, \\ 0 & \text{otherwise} \end{cases} \tag{2}$$

with $-\psi(\theta) = \theta_{(1,1,\ldots,1)}$, and the sample space $\Omega$ is a poset (partially ordered set) with respect to the partial order "$\leq$" with the least element $\bot = (1,1,\ldots,1)$. Therefore our model belongs to the *log-linear model on posets* introduced by Sugiyama et al. (2016, 2017), which is an extension of the information geometric hierarchical log-linear model (Amari, 2001; Nakahara and Amari, 2002). Each entry $q_v$ and the parameters $(\theta_v)_{v \in \Omega^+}$ in Equation (2) directly correspond to those in Equation (8) in Sugiyama et al. (2017). According to Theorem 2 in Sugiyama et al. (2017), if we introduce $(\eta_v)_{v \in \Omega^+}$ such that

$$\eta_v = \sum_{u \in \uparrow v} q_u = \sum_{u \in \Omega} \zeta(v,u)q_u, \quad \uparrow v = \{\, u \in \Omega \mid u \geq v \,\}, \tag{3}$$

for each $v \in \Omega^+$ (see Figure 1[b]), the pair $(\,(\theta_v)_{v \in \Omega^+}, (\eta_v)_{v \in \Omega^+}\,)$ is always a *dual coordinate system* of the set of normalized tensors $\boldsymbol{S} = \{\mathcal{P} \mid 0 < p_v < 1 \text{ and } \sum_{v \in \Omega} p_v = 1\}$ with respect to the sample space $\Omega$, as they are connected via *Legendre transformation*. Hence $\boldsymbol{S}$ becomes a *dually flat manifold* (Amari, 2009).

Here we formulate Legendre decomposition as a *projection* of a tensor onto a submanifold. Suppose that $B \subseteq \Omega^+$ and let $\boldsymbol{S}_B$ be the submanifold such that

$$\boldsymbol{S}_B = \{\, \mathcal{Q} \in \boldsymbol{S} \mid \theta_v = 0 \text{ for all } v \in \Omega^+ \setminus B \,\},$$

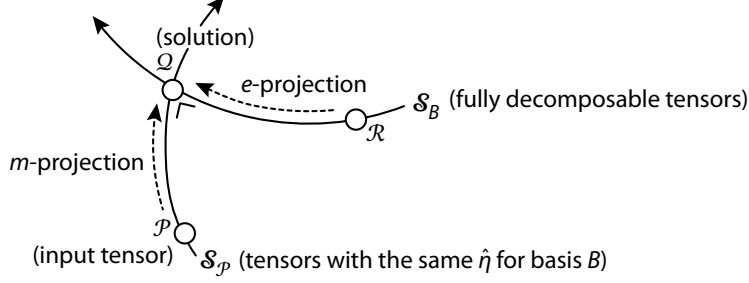

Figure 2: Projection in statistical manifold.

which is the set of fully decomposable tensors with $B$ and is an *e-flat* submanifold as it has constraints on the $\theta$ coordinate (Amari, 2016, Chapter 2.4). Furthermore, we introduce another submanifold $\boldsymbol{S}_{\mathcal{P}}$ for a tensor $\mathcal{P} \in \boldsymbol{S}$ and $A \subseteq \Omega^+$ such that

$$\boldsymbol{S}_{\mathcal{P}} = \{ \mathcal{Q} \in \boldsymbol{S} \mid \eta_v = \hat{\eta}_v \text{ for all } v \in A \},$$

where $\hat{\eta}_v$ is given by Equation (3) by replacing $q_u$ with $p_u$, which is an *m-flat* submanifold as it has constraints on the $\eta$ coordinate.

The dually flat structure of $\boldsymbol{S}$ with the dual coordinate systems $( (\theta_v)_{v \in \Omega^+}, (\eta_v)_{v \in \Omega^+} )$ leads to the following strong property: If $A = B$, that is, $(\Omega^+ \setminus B) \cup A = \Omega^+$ and $(\Omega^+ \setminus B) \cap A = \emptyset$, the intersection $\boldsymbol{S}_B \cap \boldsymbol{S}_{\mathcal{P}}$ is always a singleton; that is, the tensor $\mathcal{Q}$ such that $\mathcal{Q} \in \boldsymbol{S}_B \cap \boldsymbol{S}_{\mathcal{P}}$ always uniquely exists, and $\mathcal{Q}$ is the *minimizer* of the KL divergence from $\mathcal{P}$ (Amari, 2009, Theorem 3):

$$\mathcal{Q} = \operatorname*{argmin}_{\mathcal{R} \in \boldsymbol{S}_B} D_{\mathrm{KL}}(\mathcal{P}, \mathcal{R}). \tag{4}$$

The transition from $\mathcal{P}$ to $\mathcal{Q}$ is called the *m-projection* of $\mathcal{P}$ onto $\boldsymbol{S}_B$, and Legendre decomposition coincides with $m$-projection (Figure 2). In contrast, if some fully decomposable tensor $\mathcal{R} \in \boldsymbol{S}_B$ is given, finding the intersection $\mathcal{Q} \in \boldsymbol{S}_B \cap \boldsymbol{S}_{\mathcal{P}}$ is called the *e-projection* of $\mathcal{R}$ onto $\boldsymbol{S}_{\mathcal{P}}$. In practice, we use $e$-projection because the number of parameters to be optimized is $|B|$ in $e$-projection while it is $|\Omega \setminus B|$ in $m$-projection, and $|B| \leq |\Omega \setminus B|$ usually holds.

The $e$-projection is always convex optimization as the $e$-flat submanifold $\boldsymbol{S}_B$ is convex with respect to $(\theta_v)_{v \in \Omega^+}$. More precisely,

$$D_{\mathrm{KL}}(\mathcal{P}, \mathcal{Q}) = \sum_{v \in \Omega} p_v \log \frac{p_v}{q_v} = \sum_{v \in \Omega} p_v \log p_v - \sum_{v \in \Omega} p_v \log q_v = -\sum_{v \in \Omega} p_v \log q_v - H(\mathcal{P}),$$

where $H(\mathcal{P})$ is the entropy of $\mathcal{P}$ and independent of $(\theta_v)_{v \in \Omega^+}$. Since we have

$$-\sum_{v \in \Omega} p_v \log q_v = \sum_{v \in \Omega} p_v \left( \sum_{u \in B} \zeta(u, v)(-\theta_u) + \psi(\theta) \right), \quad \psi(\theta) = \log \sum_{v \in \Omega} \exp \left( \sum_{u \in B} \zeta(u, v)\theta(u) \right),$$

$\psi(\theta)$ is convex and $D_{\mathrm{KL}}(\mathcal{P}, \mathcal{Q})$ is also convex with respect to $(\theta_v)_{v \in \Omega^+}$.

### 2.3 Algorithm

Here we present our two gradient-based optimization algorithms to solve the KL divergence minimization problem in Equation (4). Since the KL divergence $D_{\mathrm{KL}}(\mathcal{P}, \mathcal{Q})$ is convex with respect to each $\theta_v$, the standard *gradient descent* shown in Algorithm 1 can always find the global optimum, where $\varepsilon > 0$ is a learning rate. Starting with some initial parameter set $(\theta_v)_{v \in B}$, the algorithm iteratively updates the set until convergence. The gradient of $\theta_w$ for each $w \in B$ is obtained as

$$\frac{\partial}{\partial \theta_w} D_{\mathrm{KL}}(\mathcal{P}, \mathcal{Q}) = -\frac{\partial}{\partial \theta_w} \sum_{v \in \Omega} p_v \log q_v = -\frac{\partial}{\partial \theta_w} \sum_{v \in \Omega} p_v \sum_{u \in B} \zeta(u, v)\theta_u + \frac{\partial}{\partial \theta_w} \sum_{v \in \Omega} p_v \psi(\theta)$$

$$= -\sum_{v \in \Omega} p_v \zeta(w, v) + \frac{\partial \psi(\theta)}{\partial \theta_w} = \eta_w - \hat{\eta}_w,$$

---
**Algorithm 1:** Legendre decomposition by gradient descent
---
**1** GRADIENTDESCENT($\mathcal{P}$, $B$)
**2**      Initialize $(\theta_v)_{v \in B}$;                                            // e.g. $\theta_v = 0$ for all $v \in B$
**3**      **repeat**
**4**          **foreach** $v \in B$ **do**
**5**              Compute $\mathcal{Q}$ using the current parameter $(\theta_v)_{v \in B}$;
**6**              Compute $(\eta_v)_{v \in B}$ from $\mathcal{Q}$;
**7**              $\theta_v \leftarrow \theta_v - \varepsilon(\eta_v - \hat{\eta}_v)$;
**8**      **until** *convergence of* $(\theta_v)_{v \in B}$;

---
**Algorithm 2:** Legendre decomposition by natural gradient
---
**1** NATURALGRADIENT($\mathcal{P}$, $B$)
**2**      Initialize $(\theta_v)_{v \in B}$;                                            // e.g. $\theta_v = 0$ for all $v \in B$
**3**      **repeat**
**4**          Compute $\mathcal{Q}$ using the current parameter $(\theta_v)_{v \in B}$;
**5**          Compute $(\eta_v)_{v \in B}$ from $\mathcal{Q}$ and $\Delta\boldsymbol{\eta} \leftarrow \boldsymbol{\eta} - \hat{\boldsymbol{\eta}}$;
**6**          Compute the inverse $\mathbf{G}^{-1}$ of the Fisher information matrix $\mathbf{G}$ using Equation (5);
**7**          $\boldsymbol{\theta} \leftarrow \boldsymbol{\theta} - \mathbf{G}^{-1}\Delta\boldsymbol{\eta}$
**8**      **until** *convergence of* $(\theta_v)_{v \in B}$;

---

where the last equation uses the fact that $\partial\psi(\theta)/\partial\theta_w = \eta_w$ in Theorem 2 in Sugiyama et al. (2017). This equation also shows that the KL divergence $D_{\mathrm{KL}}(\mathcal{P}, \mathcal{Q})$ is minimized if and only if $\eta_v = \hat{\eta}_v$ for all $v \in B$. The time complexity of each iteration is $O(|\Omega||B|)$, as that of computing $\mathcal{Q}$ from $(\theta_v)_{v \in B}$ (line 5 in Algorithm 1) is $O(|\Omega||B|)$ and computing $(\eta_v)_{v \in B}$ from $\mathcal{Q}$ (line 6 in Algorithm 1) is $O(|\Omega|)$. Thus the total complexity is $O(h|\Omega||B|^2)$ with the number of iterations $h$ until convergence.

Although gradient descent is an efficient approach, in Legendre decomposition, we need to repeat "decoding" from $(\theta_v)_{v \in B}$ and "encoding" to $(\eta_v)_{v \in B}$ in each iteration, which may lead to a loss of efficiency if the number of iterations is large. To reduce the number of iterations to gain efficiency, we propose to use a *natural gradient* (Amari, 1998), which is the second-order optimization method, shown in Algorithm 2. Again, since the KL divergence $D_{\mathrm{KL}}(\mathcal{P}, \mathcal{Q})$ is convex with respect to $(\theta_v)_{v \in B}$, a natural gradient can always find the global optimum. More precisely, our natural gradient algorithm is an instance of the *Bregman algorithm* applied to a convex region, which is well known to always converge to the global solution (Censor and Lent, 1981). Let $B = \{v_1, v_2, \ldots, v_{|B|}\}$, $\boldsymbol{\theta} = (\theta_{v_1}, \ldots, \theta_{v_{|B|}})^T$, and $\boldsymbol{\eta} = (\eta_{v_1}, \ldots, \eta_{v_{|B|}})^T$. In each update of the current $\boldsymbol{\theta}$ to $\boldsymbol{\theta}_{\mathrm{next}}$, the natural gradient method uses the relationship,

$$\Delta\boldsymbol{\eta} = -\mathbf{G}\Delta\boldsymbol{\theta}, \quad \Delta\boldsymbol{\eta} = \boldsymbol{\eta} - \hat{\boldsymbol{\eta}} \text{ and } \Delta\boldsymbol{\theta} = \boldsymbol{\theta}_{\mathrm{next}} - \boldsymbol{\theta},$$

which leads to the update formula

$$\boldsymbol{\theta}_{\mathrm{next}} = \boldsymbol{\theta} - \mathbf{G}^{-1}\Delta\boldsymbol{\eta},$$

where $\mathbf{G} = (g_{uv}) \in \mathbb{R}^{|B| \times |B|}$ is the Fisher information matrix such that

$$g_{uv}(\theta) = \frac{\partial\eta_u}{\partial\theta_v} = \mathbf{E}\left[\frac{\partial\log p_w}{\partial\theta_u}\frac{\partial\log p_w}{\partial\theta_v}\right] = \sum_{w \in \Omega} \zeta(u,w)\zeta(v,w)p_w - \eta_u\eta_v \quad (5)$$

as given in Theorem 3 in Sugiyama et al. (2017). Note that natural gradient coincides with Newton's method in our case as the Fisher information matrix $\mathbf{G}$ corresponds to the (negative) Hessian matrix:

$$\frac{\partial^2}{\partial\theta_u\partial\theta_v}D_{\mathrm{KL}}(\mathcal{P}, \mathcal{Q}) = -\frac{\partial\eta_u}{\partial\theta_v} = -g_{uv}.$$

The time complexity of each iteration is $O(|\Omega||B| + |B|^3)$, where $O(|\Omega||B|)$ is needed to compute $\mathcal{Q}$ from $\theta$ and $O(|B|^3)$ to compute the inverse of $\mathbf{G}$, resulting in the total complexity $O(h'|\Omega||B| + h'|B|^3)$ with the number of iterations $h'$ until convergence.

## 2.4 Relationship to Statistical Models

We demonstrate interesting relationships between Legendre decomposition and statistical models, including the exponential family and the Boltzmann (Gibbs) distributions, and show that our decomposition method can be viewed as a generalization of Boltzmann machine learning (Ackley et al., 1985). Although the connection between tensor decomposition and graphical models has been analyzed by Chen et al. (2018); Yılmaz et al. (2011); Yılmaz and Cemgil (2012), our analysis adds a new insight as we focus on not the graphical model itself but the sample space of distributions generated by the model.

### 2.4.1 Exponential family

We show that the set of normalized tensors $\boldsymbol{S} = \{\mathcal{P} \in \mathbb{R}_{>0}^{I_1 \times I_2 \times \cdots \times I_N} \mid \sum_{v \in \Omega} p_v = 1\}$ is included in the exponential family. The exponential family is defined as

$$p(x, \boldsymbol{\theta}) = \exp\left(\sum \theta_i k_i(x) + r(x) - C(\boldsymbol{\theta})\right)$$

for natural parameters $\boldsymbol{\theta}$. Since our model in Equation (1) can be written as

$$p_v = \frac{1}{\exp(\psi(\theta))} \prod_{u \in \downarrow v} \exp(\theta_u) = \exp\left(\sum_{u \in \Omega^+} \theta_u \zeta(u, v) - \psi(\theta)\right)$$

with $\theta_u = 0$ for $u \in \Omega^+ \setminus B$, it is clearly in the exponential family, where $\zeta$ and $\psi(\theta)$ correspond to $k$ and $C(\boldsymbol{\theta})$, respectively, and $r(x) = 0$. Thus, the $(\theta_v)_{v \in B}$ used in Legendre decomposition are interpreted as natural parameters of the exponential family. Moreover, we can obtain $(\eta_v)_{v \in B}$ by taking the expectation of $\zeta(u, v)$:

$$\mathbf{E}\big[\zeta(u, v)\big] = \sum_{v \in \Omega} \zeta(u, v) p_v = \eta_u.$$

Thus Legendre decomposition of $\mathcal{P}$ is understood to find a fully decomposable $\mathcal{Q}$ that has the same expectation with $\mathcal{P}$ with respect to a basis $B$.

### 2.4.2 Boltzmann Machines

A Boltzmann machine is represented as an undirected graph $G = (V, E)$ with a vertex set $V$ and an edge set $E \subseteq V \times V$, where we assume that $V = [N] = \{1, 2, \ldots, N\}$ without loss of generality. This $V$ is the set of indices of $N$ binary variables. A Boltzmann machine $G$ defines a probability distribution $P$, where each probability of an $N$-dimensional binary vector $\boldsymbol{x} \in \{0, 1\}^N$ is given as

$$p(\boldsymbol{x}) = \frac{1}{Z(\theta)} \prod_{i \in V} \exp\left(\theta_i x_i\right) \prod_{\{i,j\} \in E} \exp\left(\theta_{ij} x_i x_j\right),$$

where $\theta_i$ is a bias, $\theta_{ij}$ is a weight, and $Z(\theta)$ is a partition function.

To translate a Boltzmann machine into our formulation, let $\Omega = \{1, 2\}^N$ and suppose that

$$B(V) = \left\{ (i_1^a, \ldots, i_N^a) \in \Omega \mid a \in V \right\}, \qquad i_l^a = \begin{cases} 2 & \text{if } l = a, \\ 1 & \text{otherwise,} \end{cases}$$

$$B(E) = \left\{ (i_1^{ab}, \ldots, i_N^{ab}) \in \Omega \mid \{a, b\} \in E \right\}, \qquad i_l^{ab} = \begin{cases} 2 & \text{if } l \in \{a, b\}, \\ 1 & \text{otherwise.} \end{cases}$$

Then it is clear that the set of probability distributions, or Gibbs distributions, that can be represented by the Boltzmann machine $G$ is exactly the same as $\boldsymbol{S}_B$ with $B = B(V) \cup B(E)$ and $\exp(\psi(\theta)) = Z(\theta)$; that is, the set of fully decomposable $N$th-order tensors defined by Equation (1) with the basis $B(V) \cup B(E)$ (Figure 3). Moreover, let a given $N$th-order tensor $\mathcal{P} \in \mathbb{R}_{\geq 0}^{2 \times 2 \times \cdots \times 2}$ be an empirical distribution estimated from data, where each $p_v$ is the probability for a binary vector $v - (1, \ldots, 1) \in \{0, 1\}^N$. The tensor $\mathcal{Q}$ obtained by Legendre decomposition with $B = B(V) \cup B(E)$ coincides with the distribution learned by the Boltzmann machine $G = (V, E)$. The condition $\eta_v = \hat{\eta}_v$ in the optimization of the Legendre decomposition corresponds to the well-known learning

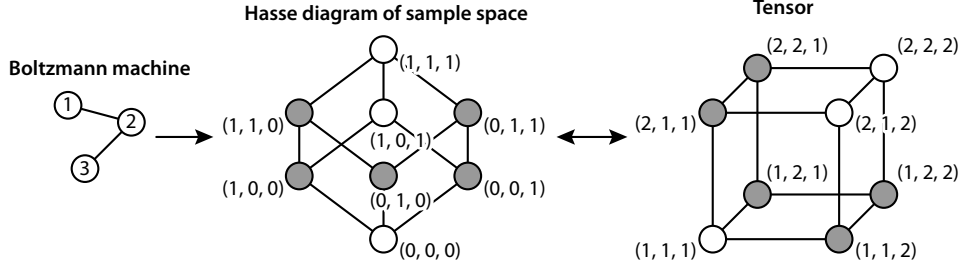

Figure 3: Boltzmann machine with $V = \{1, 2, 3\}$ and $E = \{\{1, 2\}, \{2, 3\}\}$ (left) and its sample space (center), which corresponds to a tensor (right). Grayed circles are the domain of parameters $\theta$.

equation of Boltzmann machines, where $\hat{\eta}$ and $\eta$ correspond to the expectation of the data distribution and that of the model distribution, respectively.

Therefore our Legendre decomposition is a generalization of Boltzmann machine learning in the following three aspects:

1. The domain is not limited to binary but can be ordinal; that is, $\{0, 1\}^N$ is extended to $[I_1] \times [I_2] \times \cdots \times [I_N]$ for any $I_1, I_2, \ldots, I_N \in \mathbb{N}$.
2. The basis $B$ with which parameters $\theta$ are associated is not limited to $B(V) \cup B(E)$ but can be any subset of $[I_1] \times \cdots \times [I_N]$, meaning that higher-order interactions (Sejnowski, 1986) can be included.
3. The sample space of probability distributions is not limited to $\{0, 1\}^N$ but can be any subset of $[I_1] \times [I_2] \times \cdots \times [I_N]$, which enables us to perform efficient computation by removing unnecessary entries such as missing values.

Hidden variables are often used in Boltzmann machines to increase the representation power, such as in restricted Boltzmann machines (RBMs; Smolensky, 1986; Hinton, 2002) and deep Boltzmann machines (DBMs; Salakhutdinov and Hinton, 2009, 2012). In Legendre decomposition, including a hidden variable corresponds to including an additional dimension. Hence if we include $H$ hidden variables, the fully decomposable tensor $\mathcal{Q}$ has the order of $N + H$. This is an interesting extension to our method and an ongoing research topic, but it is not a focus of this paper since our main aim is to find a lower dimensional representation of a given tensor $\mathcal{P}$.

In the learning process of Boltzmann machines, approximation techniques of the partition function $Z(\theta)$ are usually required, such as annealed importance sampling (AIS; Salakhutdinov and Murray, 2008) or variational techniques (Salakhutdinov, 2008). This requirement is because the exact computation of the partition function requires the summation over all probabilities of the sample space $\Omega$, which is always fixed to $2^V$ with the set $V$ of variables in the learning process of Boltzmann machines, and which is not tractable. Our method does not require such techniques as $\Omega$ is a subset of indices of an input tensor and the partition function can always be directly computed.

## 3 Experiments

We empirically examine the efficiency and the effectiveness of Legendre decomposition using synthetic and real-world datasets. We used Amazon Linux AMI release 2018.03 and ran all experiments on 2.3 GHz Intel Xeon CPU E5-2686 v4 with 256 GB of memory. The Legendre decomposition was implemented in C++ and compiled with `icpc` 18.0.0 [1].

Throughout the experiments, we focused on the decomposition of third-order tensors and used three types of decomposition bases in the form of $B_1 = \{v \mid i_1 = i_2 = 1\} \cup \{v \mid i_2 = i_3 = 1\} \cup \{v \mid i_1 = i_3 = 1\}$, $B_2(l) = \{v \mid i_1 = 1, i_2 \in C_2(l)\} \cup \{v \mid i_1 \in C_1(l), i_2 = 1\}$ with $C_k(l) = \{c\lfloor I_k/l \rfloor \mid c \in [l]\}$, and $B_3(l) = \{v \mid (i_1, i_2) \in H_{i_3}(l)\}$ with $H_{i_3}(l)$ being the set indices for the top $l$ elements of the $i_3$th frontal slice in terms of probability. In these bases, $B_1$ works as a normalizer for each mode, $B_2$ works as a normalizer for rows and columns of each slice, and $B_3$ highlights entries with high probabilities. We always assume that $(1, \ldots, 1)$ is not included in

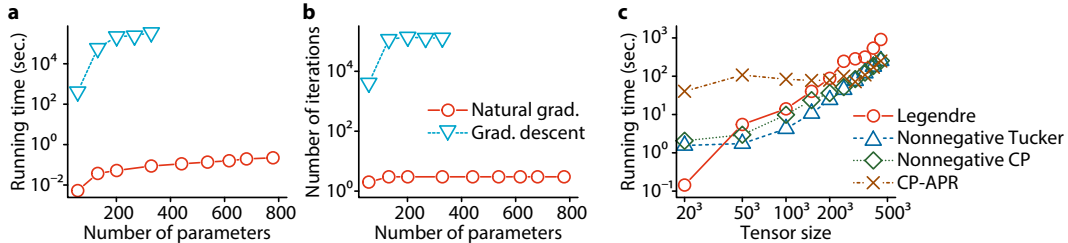

Figure 4: Experimental results on synthetic data. (**a**, **b**) Comparison of natural gradient (Algorithm 2) and gradient descent (Algorithm 1), where both algorithms produce exactly the same results. (**c**) Comparison of Legendre decomposition (natural gradient) and other tensor decomposition methods.

the above bases. The cardinality of a basis corresponds to the number of parameters used in the decomposition. We used $l$ to vary the number of parameters for decomposition in our experiments.

To examine the efficiency and the effectiveness of tensor decomposition, we compared Legendre decomposition with two standard nonnegative tensor decomposition techniques, nonnegative Tucker decomposition (Kim and Choi, 2007) and nonnegative CANDECOMP/PARAFAC (CP) decomposition (Shashua and Hazan, 2005). Since both of these methods are based on least square objective functions (Lee and Seung, 1999), we also included a variant of CP decomposition, CP-Alternating Poisson Regression (CP-APR; Chi and Kolda, 2012), which uses the KL-divergence for its objective function. We used the `TensorLy` implementation (Kossaifi et al., 2016) for the nonnegative Tucker and CP decompositions and the `tensor toolbox` (Bader et al., 2017; Bader and Kolda, 2007) for CP-APR. In the nonnegative Tucker decomposition, we always employed rank-$(m, m, m)$ Tucker decomposition with the single number $m$, and we use rank-$n$ decomposition in the nonnegative CP decomposition and CP-APR. Thus rank-$(m, m, m)$ Tucker decomposition has $(I_1 + I_2 + I_3)m + m^3$ parameters, and rank-$n$ CP decomposition and CP-APR have $(I_1 + I_2 + I_3)n$ parameters.

**Results on Synthetic Data**    First we compared our two algorithms, the gradient descent in Algorithm 1 and the natural gradient in Algorithm 2, to evaluate the efficiency of these optimization algorithms. We randomly generated a third-order tensor with the size $20 \times 20 \times 20$ from the uniform distribution and obtained the running time and the number of iterations. We set $B = B_3(l)$ and varied the number of parameters $|B|$ with increasing $l$. In Algorithm 2, we used the outer loop (from line 3 to 8) as one iteration for fair comparison and fixed the learning rate $\varepsilon = 0.1$.

Results are plotted in Figure 4(**a**, **b**) that clearly show that the natural gradient is dramatically faster than gradient descent. When the number of parameters is around $400$, the natural gradient is more than six orders of magnitude faster than gradient descent. The increased speed comes from the reduction of iterations. The natural gradient requires only two or three iterations until convergence in all cases, while gradient descent requires more than $10^5$ iterations to get the same result. In the following, we consistently use the natural gradient for Legendre decomposition.

Next we examined the scalability compared to other tensor decomposition methods. We used the same synthetic datasets and increased the tensor size from $20 \times 20 \times 20$ to $500 \times 500 \times 500$. Results are plotted in Figure 4(**c**). Legendre decomposition is slower than Tucker and CP decompositions if the tensors get larger, while the plots show that the running time of Legendre decomposition is linear with the tensor size (Figure 5). Moreover, Legendre decomposition is faster than CP-APR if the tensor size is not large.

**Results on Real Data**    Next we demonstrate the effectiveness of Legendre decomposition on real-world datasets of third-order tensors. We evaluated the quality of decomposition by the root mean squared error (RMSE) between the input and the reconstructed tensors. We also examined the scalability of our method in terms of the number of parameters.

First we examine Legendre decomposition and three competing methods on the face image dataset[2]. We picked up the first entry from the fourth mode (corresponds to lighting) from the dataset and the

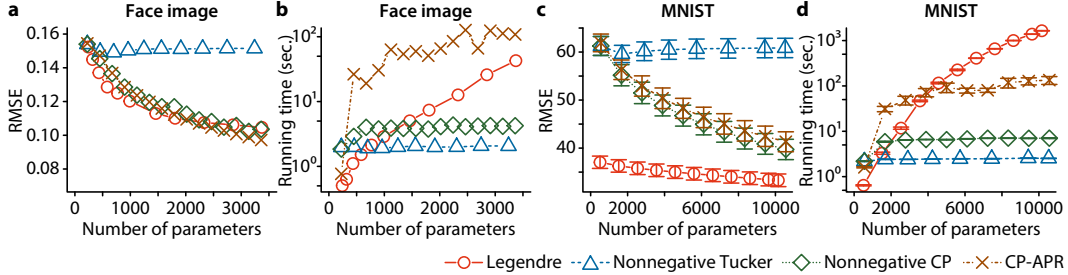

Figure 5: Experimental results on the face image dataset (**a**, **b**) and MNIST (**c**, **d**).

first 20 faces from the third mode, resulting in a single third-order tensor with a size of $92 \times 112 \times 20$, where the first two modes correspond to image pixels and the third mode to individuals. We set decomposition bases $B$ as $B = B_1 \cup B_2(l) \cup B_3(l)$. For every decomposition method, we gradually increased $l$, $m$, and $n$ and checked the performance in terms of RMSE and running time.

Results are plotted in Figure 5(**a**, **b**). In terms of RMSE, Legendre decomposition is superior to the other methods if the number of parameters is small (up to 2,000), and it is competitive with nonnegative CP decomposition and inferior to CP-APR for a larger number of parameters. The reason is that Legendre decomposition uses the information of the index order that is based on the structure of the face images (pixels); that is, rows or columns cannot be replaced with each other in the data. In terms of running time, it is slower than Tucker and CP decompositions as the number of parameters increases, while it is still faster than CP-APR.

Next we used the MNIST dataset (LeCun et al., 1998), which consists of a collection of images of handwritten digits and has been used as the standard benchmark datasets in a number of recent studies such as deep learning. We picked up the first 500 images for each digit, resulting in 10 third-order tensors with the size of $28 \times 28 \times 500$, where the first two modes correspond to image pixels. In Legendre decomposition, the decomposition basis $B$ was simply set as $B = B_3(l)$ and removed zero entries from $\Omega$. Again, for every decomposition method, we varied the number of parameters by increasing $l$, $m$, and $n$ and evaluated the performance in terms of RMSE.

Means $\pm$ standard error of the mean (SEM) across all digits from "0" to "9" are plotted in Figure 5(**c**, **d**). Results for all digits are presented in the supplementary material. Legendre decomposition clearly shows the smallest RMSE, and the difference is larger if the number of parameters is smaller. The reason is that Legendre decomposition ignores zero entries and decomposes only nonzero entries, while such decomposition is not possible for other methods. Running time shows the same trend as that of the face dataset; that is, Legendre decomposition is slower than other methods when the number of parameters increases.

# 4 Conclusion

In this paper, we have proposed Legendre decomposition, which incorporates tensor structure into information geometry. A given tensor is converted into the dual parameters $(\theta, \eta)$ connected via the Legendre transformation, and the optimization is performed in the parameter space instead of directly treating the tensors. We have theoretically shown the desired properties of the Legendre decomposition, namely, that its results are well-defined, unique, and globally optimized, in that it always finds the decomposable tensor that minimizes the KL divergence from the input tensor. We have also shown the connection between Legendre decomposition and Boltzmann machine learning.

We have experimentally shown that Legendre decomposition can more accurately reconstruct input tensors than three standard tensor decomposition methods (nonnegative Tucker decomposition, nonnegative CP decomposition, and CP-APR) using the same number of parameters. Since the shape of the decomposition basis $B$ is arbitrary, Legendre decomposition has the potential to achieve even more-accurate decomposition. For example, one can incorporate the domain knowledge into the set $B$ such that specific entries of the input tensor are known to dominate the other entries.

Our work opens the door to both further theoretical investigation of information geometric algorithms for tensor analysis and a number of practical applications such as missing value imputation.

**Acknowledgments**

This work was supported by JSPS KAKENHI Grant Numbers JP16K16115, JP16H02870, and JST, PRESTO Grant Number JPMJPR1855, Japan (M.S.); JSPS KAKENHI Grant Numbers 26120732 and 16H06570 (H.N.); and JST CREST JPMJCR1502 (K.T.).

## Footnotes

[1]Implementation is available at: `https://github.com/mahito-sugiyama/Legendre-decomposition`

[2]This dataset is originally distributed at `http://www.cl.cam.ac.uk/research/dtg/attarchive/` `facedatabase.html` and also available from the R `rTensor` package (`https://CRAN.R-project.org/` `package=rTensor`).

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
