[Supplementary Material · supplementary.pdf]

# Supplementary Material for
# Legendre Decomposition for Tensors

**Mahito Sugiyama**
National Institute of Informatics
JST, PRESTO
mahito@nii.ac.jp

**Hiroyuki Nakahara**
RIKEN Center for Brain Science
hiro@brain.riken.jp

**Koji Tsuda**
The University of Tokyo
NIMS; RIKEN AIP
tsuda@k.u-tokyo.ac.jp

Figure S1: Experimental results on each digit from 0 to 9 in MNIST. Legendre decomposition is shown in red circles, nonnegative Tucker decomposition in blue triangles, nonnegative CP decomposition in green diamonds, and CP-APR in brown crosses.