[Reviews · NeurIPS 2018]

Reviewer 1



This paper presents a non-negative tensor decomposition method, called Legendre decomposition. The objective function of Lengendre decomposition is convex and therefore, always guarantees global convergence. The relationships between lengendre decomposition to exponential family and to Boltzmann Machines are presented. Experimental results show that with the same number of parameters, Lengendre decomposition achieves smaller reconstruction error compared to existing tensor decomposition methods. This paper is generally well organized and is not difficult to follow. I consider the proposed method as novel. One important advantage of Lengendre decomposition over existing methods is that its objective function is convex and therefore always converges to global optimum. The experimental results show that the proposed method outperforms the existing methods in terms of reconstruction error. I have difficulties in understanding the concept of e-flat submanifold, m-flat submanifold, m-projection and e-projection. Therefore, I am not certain whether the formulas in Section 2.2 are correct. Technical content in the remaining parts appears to be correct. Some detailed comments are provided as follows: It looks to me that point (1) in line 34-36 is redundant provided point (2) in 36-38, because convex formulation and global convergence already imply unique solution. In line 64, it should be $v = (i_1, \ldots, i_N)$. In line 217, why do you choose the decomposition basis to be this specific form? Is it due to the structure of the datasets? Provided a dataset, how should we choose the basis? In the experiments, could you also present how the proposed method compared to other methods, in terms of running time? The proposed method consistently outperforms Nonnegative Tucker except for the experiment on digit “1” in MNIST, as shown in the supplementary material. Do you know why this happens? Is there something special in this dataset?

Reviewer 2



In the paper titled "Legendre Tensor Decomposition", the authors proposed a technique named Legendre decomposition which combines tensor structure with information geometry. Under this technique, a tensor is converted into two parameters \theta and \eta with the Legendre transformation. Some theoretical results are given, such as the uniqueness and the optimality, and the connection with Boltzmann machines. Empirical results are also quite nice. It seems the current paper did not mention much about scalability and computational complexity. If possible, could the authors provide? Overall, the paper is well-written and provides sufficient technical contribution.

Reviewer 3



Main ideas of the submission The manuscript presents an approximation of nonnegative multi-way tensorial data (or high-order probability mass functions) based on structured energy function form that minimizes the Kullback-Leibler divergence. Comparing against other multilinear decomposition methods of nonnegative tensors, the proposal approach operates on multiplicative parameters under convex objective function and converges to a globally optimal solution. It also shows interesting connections with graphical models such as the high-order Boltzmann machines. Two optimization algorithms are developed, based upon gradient and natural gradient, respectively. The experiment shows that under the same number of parameters, the proposed approach yields smaller RMSEs than the other two baseline non-negative tensor decomposition methods. Although the authors frame the proposed approach under tensor decomposition and draw a connection to statistical models, I found that it is easier to interpret it from the perspective of graphical models. Basically, the main technical point seems to be the incorporation of partial ordering constraints on the inclusion of interaction terms such that the model complexity of higher-order Boltzmann machines is under control. It seems the tensor decomposition view tends to overcomplicate the idea, and I’m not quite convinced by its benefits. Meanwhile, the experiments seem problematic and not compelling. First, the capacity of tensor models not only depends on the number of parameters but also on the operations on them. Usually, the low-dimensional components could also be interpretable, and able to capture certain underlying structure of data (e.g., multi-linear). Moreover, we do not always want an accurate reconstruction of the observed tensor (e.g., denoising). In this manuscript, the possible interactions are restricted by a partial order that looks quite artificial. In practice, would this ordering naturally arise? What kind of characteristic of the multi-variate density do we expect the Legendre decomposition to capture? Second, it seems the two baseline methods considered in the experiments are both based on least-square (LS) objective functions and multiplicative updates (Lee and Seung, 1999), while this manuscript adopts KL divergence. The LS objective may not be appropriate for non-negative smaller values. In addition, those algorithms could converge to a non-KKT point (Gonzalez and Zhang, 2015). Therefore, the worse performance of competitors could either be attributed to the objective function or the multiplicative updates. Without additional experimental evidence, it is hard to say whether the proposed method truly outperforms the competitors in terms of model capacity. To make a fair comparison, there exists KL divergence based nonnegative tensor decomposition methods, such as CP-APR (Chi and Kolda, 2012), an important baseline missing in this paper. The faster convergence of the natural gradient algorithm than its gradient counterparts is not too surprising. On page 3 line 83, what is the main reason for using the forward KL divergence KL(P, Q)? How would the zero-avoiding property affect the results in tensor reconstruction? How does it compare to the variational Bayes methods for Boltzmann machines? The is certainly room for improvements in the clarity and self-containess of this paper. I found the notations in Section 2.1 and 2.2 particularly hard to follow. Also on page 8 line 267-268, the author claims that the “Legendre decomposition effectively uses the index order information”. I’m having trouble understanding what is exactly index order information and how does it improve the reconstruction. Reference Lee, Daniel D., and H. Sebastian Seung. "Learning the parts of objects by non-negative matrix factorization." Nature401.6755 (1999): 788. Chi, Eric C., and Tamara G. Kolda. "On tensors, sparsity, and nonnegative factorizations." SIAM Journal on Matrix Analysis and Applications 33.4 (2012): 1272-1299. Gonzalez, Edward F., and Yin Zhang. Accelerating the Lee-Seung algorithm for nonnegative matrix factorization. 2005. [After Rebuttal] The CP-APR comparison included in the rebuttal serves as a more genuine competitor than the other two. The experimental results provide additional evidence on the margin of improvements (despite small) of the proposed approach. Accordingly, I will raise my rating to “weak accept”. The connection between tensor formulation and graphical models looks quite interesting, and now we know that the partial ordering constraint is at least useful for image data.

Reviewer 4



The paper is well written. It is, however, dense in many places, which makes the reading hard at the first glance. A supplementary that introduces key concepts on Legendre decomposition. The interesting idea is to compute a unique decomposition of a non-negative input tensor with a given choice of the parameter basis. Although the tensor notion is not really used in the algorithm, it does well in the limited numerical experiments. To this end, I consider the work a good contribution.

Reviewer 5



Strengths: 1. Theoretically sound tensor decomposition in terms of reconstructable tensors and optimizing the parameters an interesting approach that also seems to enjoy theoretical properties. 2. The relations to Boltzmann machines is a really interesting connection that links this method to a seemingly distant graphical model. 3. The implementation of the algorithm is also quite simple and seemingly reproducible (i.e., no randomness). 4. Despite the technical details, the paper is well-written and flows well. Weaknesses/questions: 1. Analyzing the scaling of the algorithm with respect to the tensor sizes (not necessarily comparing against other methods) on even toy examples would have been nice to see. 2. Is 256GB of memory necessary for utilizing this method? 3. Are there cases where Legendre decomposition might be less preferred compared to other methods? 4. Can this be applied to higher order (>2) tensors? 5. Often times for various applications, the rank of the decomposition can be used to control the complexity/low-rankness of the tensor intentionally (i.e., as a hyper-parameter). Can Legendre decomposition incorporate such similar feature? Final thoughts: Methods for nonnegative tensor factorization problem itself has been known to be useful but scarcely developed. This paper could be a valuable alternative for already existing methods while providing more sound theoretical properties. Also, by construction, the method makes connections to the Boltzmann machine (although it is more of an observation). The quality of the paper is overall decent, but some questions from more broad aspects (listed in weaknesses/questions above) could be addressed to further clarifications.